# Does Food Expenditure Decrease after Retirement, and for Whom?

**Ayal Kimhi** [1,2,*] and **Maya Sender** [3]

1   Department of Environmental Economics and Management, Institute of Environmental Sciences,
    The Robert H. Smith Faculty of Agriculture, Food and Environment, The Hebrew University of Jerusalem,
    Rehovot 7610001, Israel
2   Shoresh Institution for Socioeconomic Research, Kochav Yair 4486400, Israel
3   Elbit Systems, Haifa 3100401, Israel; maya.sender@elbitsystems.com
*   Correspondence: ayal.kimhi@mail.huji.ac.il

**Abstract:** This paper examines the decline in food expenditure after retirement by quantiles of the consumption distribution, by gender, and by pre-retirement employment status. The decline in food expenditure after retirement is smaller among those who were employees than among those who were self-employed, but only for females. Males who did not work did not experience a decline in food expenditure when they crossed the official retirement age, while females who did not work decreased their food expenditure in parts of the consumption distribution. These results are consistent with the two common explanations of the decline in consumption after retirement: inadequate savings and substitution of time for money. Public policy should target the inadequate savings phenomenon in order to make food consumption more sustainable during retirement.

**Keywords:** food expenditure; retirement; sustainability





## 1. Introduction

The population of developed countries, including Israel, is becoming older as a result of increased longevity and decreased fertility. Concurrently, the standard of living of the elderly population is receiving more public attention. Retirement is a critical point in the life cycle that is most relevant for well-being, because after retirement, individuals and families experience a change in their income portfolio. According to the classical life-cycle model, consumption is not affected by expected income changes, and post-retirement income changes are to a large extent expected [1], so if this is true, consumption should not be affected by retirement. Hence, if consumption declines after retirement, as has been found in many studies (the "retirement consumption puzzle"), it may be due to liquidity constraints that lead to sub-optimal savings, to unplanned or forced retirement, to uncertainty about post-retirement income or consumption, or to inadequate financial planning. Moreover, in an augmented life-cycle model, a decline in consumption does not necessarily reduce utility, because it could be that time is substituted for purchased goods either as leisure or as an input in home production [2,3]. Whatever the reason may be, the post-retirement decline in consumption deserves public attention and perhaps policy responses, because modern societies do not tolerate poverty among the elderly.

Food consumption is especially important, as its decline may lead to poor nutrition and health deprivation, and even food insecurity [4]. Food consumption is particularly important for the elderly due to its health consequences [5]. Inadequate consumption of food or consumption of unhealthy foods may speed the health deterioration of elderly people and lead to unexpectedly higher medical expenditures that could lead, in turn, to further cuts in consumption. In other words, if food consumption falls substantially after retirement, one may conclude that the pre-retirement food consumption was not sustainable.

Israel is one of the most unequal developed countries [6]. Poverty among the elderly is also quite high, and in 2020 it was ranked 13th among 37 OECD countries (Figure 1). The purpose of this research was to examine the sustainability of food consumption among the elderly in Israel. We achieved this by investigating whether the retirement consumption puzzle exists in Israel in the context of food expenditure, quantified it, and studied its determinants. We used data in synthetic cohorts constructed from consecutive household expenditure surveys and estimated the decline in food consumption after retirement. We allowed for heterogeneous responses of food expenditures to retirement by using quantile regression techniques. We also allowed for heterogeneous responses by gender, by pre-retirement employment status, and by age. Heterogeneity is especially important for policy purposes, since policy responses, if necessary, should focus on those population groups that are most vulnerable to this phenomenon.

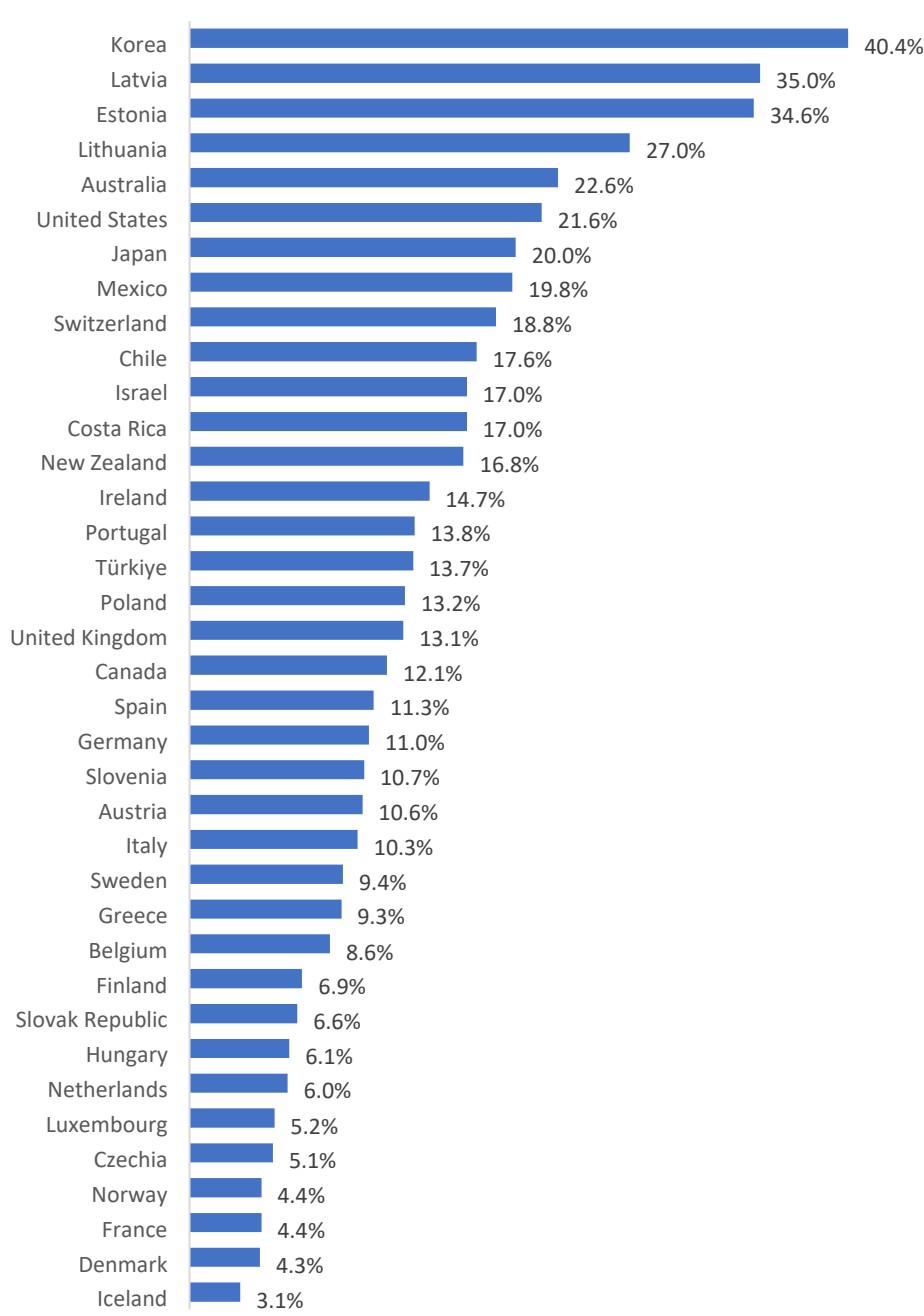

**Figure 1.** Elderly poverty rates in OECD countries, 2020 or latest.

## 2. Literature Review

Many empirical studies were able to identify between a 4% and 20% drop in consumption after retirement in different countries. Hamermesh [7] explained that some households simply do not save enough for retirement. Blake [8] found that the drop in consumption is stronger as workers rely more on private rather than public pensions. Dilnot, Disney, and Johnson [9] suggested that individuals over-estimate their post-retirement income, and this leads to sub-optimal savings. Banks, Blundell, and Tanner [10] suggested that work-related expenditures such as clothing and transportation drop after retirement, but found that this cannot explain the entire drop in overall consumption. They also suggested that people are exposed to new information about medical expenditures after they retire, because their social networks change in the direction of including older people, and this leads to higher post-retirement precautionary savings. Miniaci, Monfardini, and Weber [11] supported, using Italian data, the suggestion that work-related expenditures drop after retirement, but rejected the explanation based on over-estimation of post-retirement income. Battistin et al. [12] also showed that work-related expenditures drop after retirement, and also showed that most of the drop in consumption is due to the drop in the number of children living with their retired parents.

On the other hand, Ameriks, Caplin, and Leahy [13] found that households actually expect their consumption to drop after retirement and that their expectations are pretty much correct on average. Some households expect, though, that their consumption will not drop and even increase. Aguiar and Hurst [14] found that while work-related expenditures and food expenditures declined after retirement, leisure-related expenditures such as entertainment and charity contributions increased.

Borella, Moscarola, and Rossi [15] differentiated between voluntary and involuntary retirement. They also differentiated between retirees with different levels of education and wealth. They found that consumption declined by about 4% after retirement in Italy for both voluntary and involuntary retirees, but retirees with high levels of education and wealth did not experience a decline. When the interaction of wealth and education was investigated, it was found that consumption dropped by 8% for retirees with low levels of education and wealth; retirees with low education and high wealth did not experience a drop in consumption; and those with high levels of education and low wealth lost 10% of consumption after retirement, but only when retirement was involuntary. These results indicate that the drop in consumption after retirement is not homogeneous.

Bernheim, Skinner, and Weinberg [16] found that post-retirement consumption declined more among households that saved less, and in particular among households that had lower access to pension and social security payments. Hurd and Rohwedder [17] found that post-retirement consumption remained unchanged or even increased for households in the upper half of the wealth distribution, while it declined for households in the lower half of the wealth distribution. Fisher and Marchand [18] examined the changes in consumption after retirement along the distribution of pre-retirement consumption and found that a drop in consumption occurred only at the upper part of the distribution, and it increased with pre-retirement consumption. This implies that consumption, and perhaps also social welfare, becomes less unequal after retirement.

The studies mentioned above looked at retirement of the household head alone as the trigger for the change in consumption. The family context was examined by Lundberg, Startz, and Stillman [19], who found that the drop in consumption after retirement was significant only for married couples. They explained that women expect to live longer than their husbands and hence they have an incentive to reduce household expenditures while their husbands are alive, and they are able to do so because their husbands' bargaining power declines after they retire. Geyer et al. [20] examined the effect of a change in the legal retirement age of women in Germany and found no effect on household expenditure.

In the context of food expenditure, Allais, Leroy, and Mink [21] found that both food expenditure and consumption decline after retirement, raising concerns about the dietary balance of the elderly, especially because the decline is more pronounced among lower-

income people. Smith [22] found that it declined significantly after retirement only when retirement is involuntary and forced by health problems or disability, and when the retirees are less educated. Within this group, the decline in food expenditure is stronger for those who are not eligible for occupational pensions. Aguiar and Hurst [23] showed that the decline in food expenditure does not mean buying less food, but rather spending more time on buying more wisely. This was also the conclusion of Chen et al. [24], who found that food expenditure by retired males declined by about half after retirement, but the quantity of calories consumed remained the same. Hurd and Rohwedder [17] suggested that more time is spent on home production after retirement, replacing purchased goods. Kyureghia and Soler [25] found that elderly consumers with more time and less monetary resources spend more time in strategic shopping and home production, thereby achieving lower food prices relative to consumers in younger age groups. Smed, Normann Rønnow, and Tetens [26] found that food consumption did not decline after retirement, while the healthiness of diets increased.

Moreau and Stancaneli [27] found quantitatively and statistically significant declines in food expenditures of couples after the husband retired, but food expenditure declined only when the wife was still working. They explained that non-working wives devoted more time to household production and hence their food expenditures were lower even before their husbands' retirement. Kimhi and Itin-Shwartz [28] found that in dual-income households, the husband's retirement reduced food expenditure, while the wife's retirement had no significant effect. In single-income households, the negative effect of the husband's retirement disappeared. This may be due to the changing roles of husbands in home production after retirement in dual-income households, but not in single-income households. They also found that food expenditure declined after retirement for single males, but not for single females. A plausible explanation of their results is that the decline in food expenditures after retirement is mainly due to increased home production of meals, thereby reducing the monetary cost of meals. This is supported by Bonsang and Van Soest [29], who showed that the transition from work to retirement significantly increased the time spent on home production.

To summarize, the existing literature examined the retirement-consumption puzzle from many different angles using many different empirical strategies. The literature that focused on food expenditures is much less developed, though. The purpose of this study was to adopt the most suitable empirical strategies from the general literature (and developing them further) for studying the changes in food expenditures after retirement.

## 3. Empirical Methodology

This research adopted the empirical approach of Fisher and Marchand [18]. The idea is to use repeated cross-sectional data to create pseudo-cohorts of individuals born in the same year and to follow them over time, before and after retirement. While Fisher and Marchand [18] focused on the retirement of males only, we considered the retirement of both males and females. An individual will be defined as retired if he/she is above the official retirement age (which is different for males and females and has increased over time) and has not worked in the previous three months. The basic equation that is estimated is the following:

$$\text{Ln}(C) = \alpha + \beta R + X\gamma + \varepsilon \tag{1}$$

where C is monthly expenditure (either total expenditure or food expenditure) per equivalent adult (measured in constant prices), R is a binary indicator of retirement status, and X is a matrix of explanatory variables including cohort dummies. Our focus is on the estimated $\beta$ coefficient, which measures the correlation between expenditure and retirement status. It should be emphasized that this correlation cannot be interpreted as a causal relationship due to the potential endogeneity of the retirement status. This stems from the fact that the timing of retirement may be influenced by individuals' desired consumption [28].

Next, we want to allow heterogeneity of the changes in consumption with respect to the pre-retirement employment status. Specifically, we want to estimate separate retirement

coefficients for people who were wage employees, self-employed, or not employed before retirement. In order to do that we augment Equation (1) as:

$$Ln(C) = \alpha + \beta_1 Salaried + \beta_2 Self\text{-}employed + \beta_3 Not\text{-}working + X\gamma + \varepsilon \qquad (2)$$

Note that the β coefficients of Equation (2) are expected to be of opposite sign to the β coefficients of Equation (1), because they now show by how much consumption was larger before retirement compared to after retirement.

The β coefficient measures how consumption changes with retirement at the mean of the consumption distribution. Given the immense literature showing heterogeneity in this change in many dimensions, it makes sense to allow heterogeneity with respect to pre-retirement consumption as well. In fact, Aguila, Attanasio, and Meghir [30] found that consumption decreases after retirement for low-consumption households and increases for high-consumption households. We estimated the change in consumption after retirement for each percentile of the consumption distribution using quantile regression methodology [31–34]. For this purpose, Equation (1) can be estimated separately for each quantile q of the distribution, yielding quantile-specific coefficients $\beta_q$, as in Equation (3):

$$Ln(C) = \alpha_q + \beta_q R + X\gamma_q + \varepsilon \qquad (3)$$

In practice, the equations for all quantiles are estimated simultaneously using Stata's "sqreg" command. The advantage of the simultaneous estimation is that it provides more precise estimates of the standard errors of the coefficients and enables testing the equality of the coefficients related to different quantiles.

We now want to allow for heterogeneity of the change in food expenditure by pre-retirement labor force status, as in Equation (2), and also by gender. For this purpose, we combine Equations (2) and (3) and augment the model with interactions between gender and labor market status, as in Equation (4):

$$\begin{aligned} Ln(C) = \alpha + \beta_{1q} Male \times Salaried + \beta_{2q} Male \times Self\text{-}employed + \\ \beta_{3q} Male \times Not\text{-}working + \beta_{4q} Female \times Salaried + \beta_{5q} Female \times Self\text{-}employed + \\ \beta_{6q} Female \times Not\text{-}working + X\gamma + \varepsilon \end{aligned} \qquad (4)$$

## 4. Data

The data for this research were obtained from Household Expenditure Surveys in Israel for the years 1997–2012. Observations with zero income or expenditures were excluded, as well as residents of East Jerusalem, because East Jerusalem was not surveyed in all rounds due to security constraints. Every two consecutive surveys were merged in order to guarantee a sufficient number of observations in each cohort. For example, the youngest cohort we defined included individuals who were 50–51 years old in 1997 or in 1998. This cohort was followed until they were 64–65 years old in 2011 or 2012, for a total of eight 2-year periods. For this cohort, only females had passed retirement age during that time span. The oldest cohort included individuals who were 58–59 years old in 1997 or 1998 and 72–73 years old in 2011 or 2012. Including the cohorts in between, we used a total of five cohorts. Younger and older cohorts were excluded from the analysis because they were not observed both before and after the official retirement age. Table 1 summarizes the definitions of the cohorts.

An individual was classified as retired if he or she was above the official retirement age and had not worked in the last three months. It should be noted that the official retirement age increased over the sample period. The retirement age of males (females) was 65 (60) up to 2005, 66 (61) between 2005 and 2009, and 67 (62) since 2009. The fraction of retirees, according to the definition above, is 30% in the entire sample, and it ranges from 26% among the 60–61 age group to 87% among the 72–73 age group.

**Table 1.** List of cohorts and number of observations.

| Age | Cohort A | Cohort B | Cohort C | Cohort D | Cohort E | Sample Size |
|---|---|---|---|---|---|---|
| 50–51 | 1997–1998 | | | | | 906 |
| 52–53 | 1999–2000 | 1997–1998 | | | | 1586 |
| 54–55 | 2001–2002 | 1999–2000 | 1997–1998 | | | 1995 |
| 56–57 | 2003–2004 | 2001–2002 | 1999–2000 | 1997–1998 | | 2564 |
| 58–59 | 2005–2006 | 2003–2004 | 2001–2002 | 1999–2000 | 1997–1998 | 3002 |
| 60–61 | 2007–2008 | 2005–2006 | 2003–2004 | 2001–2002 | 1999–2000 | 2978 |
| 62–63 | 2009–2010 | 2007–2008 | 2005–2006 | 2003–2004 | 2001–2002 | 2901 |
| 64–65 | 2011–2012 | 2009–2010 | 2007–2008 | 2005–2006 | 2003–2004 | 2883 |
| 66–67 | | 2011–2012 | 2009–2010 | 2007–2008 | 2005–2006 | 2099 |
| 68–69 | | | 2011–2012 | 2009–2010 | 2007–2008 | 1504 |
| 70–71 | | | | 2011–2012 | 2009–2010 | 1013 |
| 72–73 | | | | | 2011–2012 | 544 |
| Sample size | 6785 | 4977 | 3889 | 4059 | 4265 | 23,975 |

Expenditures and income were measured as monthly averages in the previous three months and were expressed in 2012 prices. Standardized income and expenditure variables were obtained by dividing income and consumption by an age-adjusted standardized measure of household size to adjust for economies of scale in household expenditures. The standardization scheme is presented in Appendix A.

Table 2 compares the expenditures of retirees and non-retirees. Total expenditures of retirees were (unconditionally) 7% lower than that of non-retirees, and this was within the range of the estimates obtained in the literature. However, much of the difference was due to the lower expenditures on education, culture, and entertainment, as well as transportation and communication. It is likely that money expenditures on these items are substituted by time among retirees. On the other hand, food, housing, and health expenditures were higher among retirees by 3%, 11%, and 18%, respectively. Expenditures on meals outside of the home were roughly the same.

**Table 2.** Average expenditures among retirees and non-retirees (NIS per month).

| Variable | Retiree | Non-Retiree |
|---|---|---|
| Total expenditures * | 5657 | 6074 |
| Food * | 916 | 892 |
| Meals outside of home | 202 | 207 |
| Housing * | 1626 | 1437 |
| Home maintenance | 591 | 596 |
| Furniture and home equipment * | 307 | 352 |
| Clothing and shoes | 236 | 240 |
| Health * | 482 | 409 |
| Education, culture and entertainment * | 512 | 692 |
| Transportation and communication * | 1026 | 1326 |
| Other goods and services * | 285 | 304 |

* Difference statistically significant at 1%.

The higher health expenditures among retirees are logically accepted. The higher housing expenditures may be because household size is smaller among retirees, and they live in a house that is larger than what they need. We have no logical explanation for the higher food expenditure among retirees. It now remains to be seen if these differences still hold after controlling for a set of socio-economic attributes.

Table 3 shows the explanatory variables used in the regression analysis as well as their means among retirees and non-retirees. All mean differences are significantly different from zero. It can be seen that retirees are predominantly women, because of their lower official retirement age and higher life expectancy. Retirees are also older, on average, and fewer of them are still married (probably due more to widowhood rather than divorce). Retirees are

more educated, in part because they are older and in part because more educated workers tend to prolong their working career even beyond the official retirement age. Retirees have fewer rooms in their house, reflecting their smaller household size. Only 9% of the retirees' households have more than one car, compared to 22% of the non-retirees. This is likely a result of their smaller household size and the fact that they do not have to drive to work. The labor income of retirees' households is much smaller, while their non-labor income is much higher. Table 3 also shows that the share of retirees is larger in the older cohorts because older cohorts are observed at older ages (Table 1).

**Table 3.** Explanatory variables and their sample means.

| Variable | Retiree | Non-Retiree |
|---|---|---|
| Male | 0.25 | 0.56 |
| Age | 66.13 | 58.61 |
| Married | 0.69 | 0.82 |
| Years of schooling (non-Haredi) * | 10.78 | 12.45 |
| Years of schooling (Haredi) * | 0.11 | 0.26 |
| Center | 0.44 | 0.46 |
| Jewish | 0.87 | 0.91 |
| Rooms | 3.69 | 4.06 |
| More than one car | 0.09 | 0.22 |
| Labor income | 1414.66 | 6132.10 |
| Non-labor income | 5163.29 | 3234.04 |
| Cohort A | 0.09 | 0.36 |
| Cohort B | 0.11 | 0.24 |
| Cohort C | 0.17 | 0.15 |
| Cohort D | 0.25 | 0.13 |
| Cohort E | 0.35 | 0.10 |

* Haredi (ultra-orthodox) schooling is very different in terms of curriculum and labor market impact [35], hence the variables were separated. The sample means include zeros for those who are not in the group.

## 5. Results

Equation (1) was estimated by OLS for each expenditure item. Table 4 shows the coefficient of retirement status, indicating the percentage change in consumption after retirement controlling for the other explanatory variables listed in Table 3 (the full regression results are in Appendix B). It can be seen that total expenditures decline by 3.5% after retirement. Food expenditures decline but at a lower rate than total expenditures, less than 3 percent (as opposed to the results of Fisher et al. [36]), while expenditures on meals away from home do not change in a statistically significant manner. Housing and health expenditures decline by higher rates than total expenditures after retirement. Expenditures on education, culture, and entertainment increase after retirement, probably due to the complementarity of these expenditures with free time. Note that retirement status is obviously correlated with age, but we control for age in the regression in order to obtain a "clean" measure of the decline in consumption after retirement.

The estimated β coefficients of Equation (2) are shown in Table 5 (the full regression results are in Appendix C). These coefficients imply that the consumption of those who were not employed actually increased after retirement, while that of those who were working decreased even more sharply than the earlier results implied. This shows the importance of controlling for pre-retirement employment status. Note that those who were not employed prior to retirement were classified as retired after crossing the official retirement age. Compared to post-retirement, food expenditures were higher before retirement by 6% among the salaried workers and by 6.8% among the self-employed, while they were even lower among those who did not work. Similarly, meals outside of the home were higher by 9.2% before retirement among salaried employees and by almost 20 percent among the self-employed. Housing expenditures also declined the most among those who were working before retirement, and the same is true for health expenditures, home maintenance, furniture, and home equipment. Altogether, the decline of food expenditure after retirement

was twice as large as the decline that was estimated without conditioning on pre-retirement employment status, but it was moderate compared to other expenditures, testifying to the classification of food as a necessity.

**Table 4.** Percentage change in consumption after retirement, by expenditure item.

| Variable | Coefficient | Standard Error |
|---|---|---|
| Total expenditures | −0.035 ** | (0.009) |
| Food | −0.029 * | (0.010) |
| Meals outside of home | −0.041 | (0.036) |
| Housing | −0.114 ** | (0.009) |
| Home maintenance | −0.077 ** | (0.012) |
| Health | −0.081 ** | (0.022) |
| Education, culture, and entertainment | 0.088 ** | (0.021) |
| Clothing and shoes | −0.012 | (0.026) |
| Transportation and communication | 0.025 | (0.021) |
| Furniture and home equipment | −0.042 | (0.035) |
| Other goods and services | −0.056 * | (0.026) |

Notes: Standard errors in parentheses. * Statistically significant at 5%. ** Statistically significant at 1%.

**Table 5.** Percentage change in consumption after retirement, by expenditure item and pre-retirement status.

| Variable | $\beta_1$—Salaried | Standard Error | $\beta_2$—Self Employed | Standard Error | $\beta_3$—Not Working | Standard Error |
|---|---|---|---|---|---|---|
| Total expenditures | 0.119 ** | (0.010) | 0.108 ** | (0.014) | −0.024 ** | (0.010) |
| Food | 0.060 ** | (0.012) | 0.068 ** | (0.015) | 0.005 | (0.011) |
| Meals outside of home | 0.092 ** | (0.040) | 0.197 ** | (0.050) | −0.022 | (0.039) |
| Housing | 0.210 ** | (0.010) | 0.236 ** | (0.013) | 0.037 ** | (0.009) |
| Home maintenance | 0.139 ** | (0.013) | 0.274 * | (0.017) | 0.011 | (0.012) |
| Health | 0.163 ** | (0.025) | 0.102 ** | (0.032) | 0.025 | (0.023) |
| Education, culture, and entertainment | −0.018 | (0.024) | −0.045 ** | (0.031) | −0.136 ** | (0.022) |
| Clothing and shoes | 0.070 ** | (0.030) | 0.017 | (0.038) | −0.022 | (0.028) |
| Transportation and communication | 0.040 | (0.024) | −0.036 | (0.031) | −0.062 ** | (0.022) |
| Furniture and home equipment | 0.149 ** | (0.039) | 0.101 * | (0.051) | −0.035 | (0.037) |
| Other goods and services | 0.115 ** | (0.029) | 0.000 | (0.038) | 0.031 | (0.027) |

Notes: Standard errors in parentheses. * Statistically significant at 5%. ** Statistically significant at 1%.

The estimated coefficients of the change in food expenditure after retirement using the quantile regression (4) are reported in Table 6 (the full set of estimated coefficients is in Appendix D). It is easy to see that the decline in food consumption after retirement is not uniform throughout the consumption distribution. For salaried workers, the decline in food consumption is in most cases higher at higher levels of consumption, although it is slightly lower at the top of the distribution. The largest decline is around the 70th and the 80th percentiles. This is also true for self-employed males, while for self-employed females the decline is strongest between the 40th and the 70th percentiles. Among the non-workers, there was no significant change in the food consumption of males following retirement, while for females there was a significant decline between the 50th and the 70th percentiles. This gender difference can be due to the reliance of married females on their husbands, who may be retiring at the same time [28].

**Table 6.** Percentage change in food expenditure after retirement, by gender and quantile.

| Quantile | $\beta_1$—Salaried | Standard Error | $\beta_2$—Self Employed | Standard Error | $B_3$—Not Working | Standard Error |
|---|---|---|---|---|---|---|
| Males | | | | | | |
| P10 | 0.042 | 0.029 | 0.032 | 0.032 | −0.016 | 0.029 |
| P20 | 0.057 * | 0.028 | 0.019 | 0.034 | −0.025 | 0.026 |
| P30 | 0.079 ** | 0.021 | 0.035 | 0.026 | −0.013 | 0.02 |
| P40 | 0.078 ** | 0.021 | 0.045 | 0.024 | −0.028 | 0.019 |
| P50 | 0.077 ** | 0.023 | 0.052 | 0.026 | −0.035 | 0.021 |
| P60 | 0.081 ** | 0.019 | 0.074 ** | 0.021 | −0.02 | 0.02 |
| P70 | 0.109 ** | 0.022 | 0.109 ** | 0.023 | 0.017 | 0.02 |
| P80 | 0.109 ** | 0.025 | 0.108 ** | 0.027 | −0.004 | 0.024 |
| P90 | 0.099 ** | 0.031 | 0.126 ** | 0.035 | 0.006 | 0.03 |
| OLS | 0.082 ** | 0.017 | 0.072 ** | 0.02 | −0.011 | 0.017 |
| Females | | | | | | |
| P10 | 0.028 | 0.027 | 0.004 | 0.052 | 0.014 | 0.027 |
| P20 | 0.025 | 0.022 | 0.009 | 0.036 | 0.042 | 0.024 |
| P30 | 0.041 | 0.022 | 0.059 | 0.035 | 0.028 | 0.021 |
| P40 | 0.032 | 0.017 | 0.129 ** | 0.043 | 0.035 | 0.02 |
| P50 | 0.039 * | 0.019 | 0.133 ** | 0.028 | 0.038 * | 0.018 |
| P60 | 0.049 ** | 0.016 | 0.113 ** | 0.025 | 0.043 ** | 0.015 |
| P70 | 0.071 ** | 0.016 | 0.130 ** | 0.03 | 0.048 ** | 0.016 |
| P80 | 0.063 ** | 0.019 | 0.119 ** | 0.028 | 0.017 | 0.017 |
| P90 | 0.051 ** | 0.017 | 0.101 * | 0.043 | 0.016 | 0.02 |
| OLS | 0.043 ** | 0.014 | 0.084 ** | 0.026 | 0.031 * | 0.014 |

Notes: Standard errors in parentheses. * Statistically significant at 5%. ** Statistically significant at 1%.

For males, the decline in food consumption was not very different for those who were salaried employees before retirement and those who were self-employed, while for females, the decline among the self-employed was much larger. Among those who were salaried employees before retirement, the decline in food consumption was considerably larger for males than for females. On the contrary, among those who were self-employed, the decline was slightly larger for females. These gender differences can result from the gender-specific retirement regulation that gives female employees more flexibility to choose an optimal retirement age. While males are eligible for full retirement benefits at age 67, females can retire at age 62 and still be eligible for full benefits. These differences can also result from the gender wage differential in Israel, which is among the highest among developed countries [37].

We now return to the question of what is the reason for the decline in food expenditure after retirement—is it a natural decline because retirees spend more time shopping for cheaper food and cooking at home rather than purchasing prepared food, or simply a result of inadequate savings? The fact that the decrease in food expenditures generally increases along the consumption distribution supports the inadequate savings explanation. Given income (which we controlled for), higher consumption implies lower savings. The fact that the decline in food expenditure was larger for the self-employed, at least for females, also supports the inadequate savings explanation. This is because until recently, the self-employed were not eligible to save for retirement in pension funds and enjoy the built-in tax benefits [38], probably leading to less savings. The fact that the decrease in food expenditure is almost non-existent among those who did not work supports the explanation of substituting time for money after retirement because those who did not work did not experience a change in the value of their time. Hence, our results do not provide a definite answer to this question. Most likely, both explanations are valid.

### 6. Summary and Conclusions

This paper examined the decline in food consumption after retirement. Previous research has shown that this decline may be quite heterogeneous in the population, and in particular, could be different for different quantiles of the consumption distribution. We adopted the best-practice empirical methodology from the literature on overall consumption for the specific case of food expenditures. We extended the analysis to differentiate the decline in consumption by gender and by pre-retirement employment status and found that each extension, in turn, makes a meaningful difference. Specifically, we found that the decline in food consumption was not very different among males who were employees and among males who were self-employed, while for females the decline in consumption was larger for self-employed. Males who did not work did not experience a decline in food consumption, while for females who did not work, food consumption declined in parts of the consumption distribution. These results are consistent with the two common explanations of the "retirement consumption puzzle": inadequate savings and substitution of time for money.

While replacing time for money in food preparation after retirement is natural, the decline in food consumption after retirement due to inadequate savings requires policy responses. It is important to note that there were some likely changes in relevant parameters since our data were collected. Saving in pension funds is now mandatory for all workers in Israel, including the self-employed, and employment rates of older people have increased, perhaps due to improved health. Therefore, the inadequate savings explanation for the decline in food consumption after retirement is perhaps not as strong today as it has been in the past. However, the mandatory savings imply a relatively low replacement rate that does not allow retirees to retain their pre-retirement standard of living [38]. A possible policy response is to make mandatory pension savings progressive so that higher-income workers will have to deposit a larger percentage of their incomes. This will likely reduce the gradient in the decline of food consumption along the consumption distribution by reducing pre-retirement food consumption and increasing post-retirement food consumption among those who do not save enough for retirement. This policy will not only improve the standard of living of the elderly but could also have positive external effects by allowing for adequate food consumption and enhancing the healthfulness of the diets of the elderly, thereby saving on public health expenditures. In addition, previous research has shown that more flexible retirement arrangements make the transition to retirement smoother in terms of consumption [39]. All in all, these policies could make food consumption more sustainable during the expected but somewhat uncertain event of retirement [40–43].

Future research in this area can take several different routes. First, it has been shown [44] that the decline in consumption after retirement is not independent of household wealth in general and housing in particular, and is also affected by bequest motives. While our empirical analysis controlled for the number of rooms in the house as a proxy for housing wealth, our data do not have an intergenerational component that allows for the treatment of bequest motives. Second, a closer inspection of trends in household finances (liquid assets and debt) can shed more light on the causes of the retirement consumption puzzle [45]. Third, a decomposition of household food purchases into their nutritional components may enable an assessment of the health implications of the drop in food consumption in old age [46–48]. Fourth, looking at the joint changes in various consumption categories may shed more light on the heterogeneous changes in consumption patterns after retirement [49]. Finally, studies that used longitudinal data (for example [28,49]) have shown that such data allow for both accounting for a richer set of heterogeneous behaviors and controlling for the endogeneity of the retirement decision.

**Author Contributions:** Conceptualization, A.K.; Methodology, A.K. and M.S.; Software, M.S.; Formal analysis, A.K. and M.S.; Investigation, A.K. and M.S.; Resources, A.K.; Data curation, M.S.; Writing—original draft, A.K.; Writing—review and editing, M.S.; Supervision, A.K.; Project administration, A.K.; Funding acquisition, A.K. All authors have read and agreed to the published version of the manuscript.

**Funding:** This research was funded by the research fund of the National Insurance Institute.

**Data Availability Statement:** The data are available from the authors upon request.

**Conflicts of Interest:** Author Maya Sender was employed by the company Elbit Systems. The remaining authors declare that the research was conducted in the absence of any commercial or financial relationships that could be construed as a potential conflict of interest.

## Appendix A

**Table A1.** Computation of standardized household size.

| Number of Household Members | Standardized Household Size | Marginal Addition of Household Members |
|---|---|---|
| 1 | 1.25 | 1.25 |
| 2 | 2 | 0.75 |
| 3 | 2.65 | 0.65 |
| 4 | 3.2 | 0.55 |
| 5 | 3.75 | 0.55 |
| 6 | 4.25 | 0.5 |
| 7 | 4.75 | 0.5 |
| 8 | 5.2 | 0.45 |
| 9 | 5.6 | 0.4 |
| additional | | 0.4 |

## Appendix B

**Table A2.** Complete results of estimating Equation (1).

| Explanatory Variable | Total | Food | Meals Outside of Home | Housing | Home Maintenance | Health | Education, Culture and Entertainment | Clothing and Shoes | Transportation and Communication | Furniture and Home Equipment | Other Goods and Services |
|---|---|---|---|---|---|---|---|---|---|---|---|
| Retired | −0.035 ** | −0.029 ** | −0.041 | −0.114 ** | −0.077 ** | −0.081 ** | 0.088 ** | −0.012 | 0.025 | −0.042 | −0.056 * |
| | 0.009 | 0.01 | 0.036 | 0.009 | 0.012 | 0.022 | 0.021 | 0.026 | 0.021 | 0.035 | 0.026 |
| Male | −0.028 ** | −0.040 ** | 0.029 | −0.069 ** | −0.053 ** | −0.103 ** | 0.035* | −0.025 | 0.064 ** | −0.041 | −0.01 |
| | 0.006 | 0.007 | 0.023 | 0.006 | 0.008 | 0.015 | 0.014 | 0.017 | 0.014 | 0.024 | 0.017 |
| Age | 0.013 ** | 0.010 ** | 0.026 ** | 0.031 ** | 0.024 ** | 0.057 ** | −0.017 ** | 0.014 ** | 0 | −0.018 ** | −0.003 |
| | 0 | 0 | 0.003 | 0 | 0 | 0.001 | 0.001 | 0.002 | 0.001 | 0.003 | 0.002 |
| Married | −0.096 ** | 0.068 ** | −0.311 ** | −0.185 ** | −0.104 ** | 0.082 ** | −0.090 ** | −0.206 ** | 0.03 | −0.125 ** | −0.292 ** |
| | 0.007 | 0.008 | 0.028 | 0.007 | 0.009 | 0.018 | 0.016 | 0.021 | 0.017 | 0.028 | 0.021 |
| Years of schooling (non−Haredi) | 0.026 ** | 0.010 ** | 0.043 ** | 0.018 ** | 0.018 ** | 0.024 ** | 0.056 ** | 0.019 ** | 0.051 ** | 0.017 ** | 0.017 ** |
| | 0 | 0 | 0.002 | 0 | 0 | 0.001 | 0.001 | 0.001 | 0.001 | 0.002 | 0.002 |
| Years of schooling (Haredi) | 0.006 ** | 0.004 ** | 0 | 0.007 ** | 0.007 ** | 0.002 | 0.015 ** | 0.008 ** | 0 | 0.009* | 0.004 |
| | 0.001 | 0.001 | 0.005 | 0.001 | 0.001 | 0.002 | 0.002 | 0.003 | 0.002 | 0.004 | 0.003 |
| Center | 0.118 ** | 0.006 | 0.153 ** | 0.269 ** | 0.122 ** | 0.053 ** | 0.069 ** | 0.054 ** | 0.102 ** | −0.058 ** | 0.017 |
| | 0.006 | 0.007 | 0.022 | 0.006 | 0.007 | 0.014 | 0.013 | 0.016 | 0.013 | 0.022 | 0.016 |
| Jewish | 0 | −0.233 ** | 0.08 | 0.367 ** | −0.058 ** | 0.145 ** | 0.554 ** | −0.530 ** | 0.014 | −0.270 ** | −0.133 ** |
| | 0.011 | 0.012 | 0.045 | 0.011 | 0.013 | 0.027 | 0.026 | 0.029 | 0.025 | 0.042 | 0.03 |
| Rooms | 0.101 ** | 0.039 ** | 0.029 ** | 0.092 ** | 0.156 ** | 0.041 ** | 0.130 ** | 0.052 ** | 0.159 ** | 0.054 ** | 0.045 ** |
| | 0.002 | 0.003 | 0.009 | 0.002 | 0.003 | 0.006 | 0.006 | 0.007 | 0.006 | 0.009 | 0.007 |
| More than one car | 0.276 ** | 0.086 ** | 0.458 ** | 0.022 ** | 0.226 ** | 0.228 ** | 0.460 ** | 0.197 ** | 0.676 ** | 0.171 ** | 0.234 ** |
| | 0.008 | 0.009 | 0.027 | 0.008 | .01 | 0.019 | 0.018 | 0.021 | 0.018 | 0.029 | 0.022 |
| Log (labor income) | 0.025 ** | 0.008 ** | 0.030 ** | 0 | 0.017 ** | 0.008 ** | 0.040 ** | 0.026 ** | 0.074 ** | 0.017 ** | 0.027 ** |
| | 0 | 0.001 | 0.004 | 0 | 0.001 | 0.002 | 0.002 | 0.003 | 0.002 | 0.004 | 0.003 |

**Table A2.** *Cont.*

| Explanatory Variable | Total | Food | Meals Outside of Home | Housing | Home Maintenance | Health | Education, Culture and Entertainment | Clothing and Shoes | Transportation and Commu-nication | Furniture and Home Equipment | Other Goods and Services |
|---|---|---|---|---|---|---|---|---|---|---|---|
| Log (non−labor income) | 0.143 ** | 0.071 ** | 0.118 ** | 0.123 ** | 0.127 ** | 0.161 ** | 0.147 ** | 0.095 ** | 0.199 ** | 0.152 ** | 0.116 ** |
| | 0.002 | 0.003 | 0.009 | 0.002 | 0.003 | 0.006 | 0.006 | 0.007 | 0.006 | 0.01 | 0.007 |
| Cohort A | 0.110 ** | 0.044 ** | 0.200 ** | 0.177 ** | 0.157 ** | 0.326 ** | −0.022 | 0.192 ** | 0.152 ** | −0.156 ** | 0.003 |
| | 0.01 | 0.011 | 0.038 | 0.01 | 0.012 | 0.024 | 0.022 | 0.028 | 0.023 | 0.037 | 0.028 |
| Cohort B | 0.113 ** | 0.044 ** | 0.175 ** | 0.144 ** | 0.152 ** | 0.269 ** | 0.004 | 0.173 ** | 0.160 ** | −0.085 * | 0.047 |
| | 0.01 | 0.011 | 0.038 | 0.01 | 0.012 | 0.024 | 0.022 | 0.028 | 0.023 | 0.037 | −0.028 |
| Cohort C | 0.080 ** | 0.034 ** | 0.134 ** | 0.111 ** | 0.111 ** | 0.167 ** | 0.012 | 0.109 ** | 0.097 ** | −0.049 | 0.045 |
| | 0.01 | 0.011 | 0.039 | 0.01 | 0.012 | 0.024 | 0.023 | 0.028 | 0.023 | 0.038 | −0.028 |
| Cohort D | 0.031 ** | 0.016 | 0.023 | 0.047 ** | 0.051 ** | 0.090 ** | −0.008 | 0.032 | 0.044 * | −0.007 | −0.017 |
| | 0.01 | 0.011 | 0.039 | 0.009 | 0.012 | 0.023 | 0.022 | 0.028 | 0.022 | 0.037 | −0.027 |
| Intercept | 5.618 ** | 5.284 ** | 1.014 ** | 3.360 ** | 2.740 ** | −0.195 | 3.689 ** | 3.240 ** | 2.839 ** | 4.635 ** | 4.075 ** |
| | −0.051 | 0.056 | 0.182 | 0.05 | 0.062 | 0.12 | 0.111 | 0.137 | 0.112 | 0.184 | −0.136 |
| R−squared | 40.50% | 8.70% | 14.30% | 42.70% | 31.30% | 14.40% | 28.20% | 6.60% | 35.30% | 3.50% | 5.50% |
| Observations | 23,975 | 23,354 | 11,821 | 23,581 | 23,607 | 21,762 | 22,940 | 15,619 | 23,807 | 17,181 | 20,371 |

Notes: Standard error below coefficient. * Statistically significant at 5%. ** Statistically significant at 1%.

## Appendix C

**Table A3.** Complete results of estimating Equation (2).

| Explanatory Variable | Total | Food | Meals Outside of Home | Housing | Home Maintenance | Health | Education, Culture, and Entertainment | Clothing and Shoes | Transportation and Commu-nication | Furniture and Home Equipment | Other Goods and Services |
|---|---|---|---|---|---|---|---|---|---|---|---|
| Employee | 0.119 ** | 0.060 ** | 0.092* | 0.210 ** | 0.139 ** | 0.163 ** | −0.018 | 0.070 ** | 0.04 | 0.149 ** | 0.115 ** |
| | −0.01 | −0.012 | −0.04 | −0.01 | −0.013 | −0.025 | −0.024 | −0.03 | −0.024 | −0.039 | −0.029 |
| Self−employed | 0.108 ** | 0.068 ** | 0.197 ** | 0.236 ** | 0.274 ** | 0.102 ** | −0.045 ** | 0.017 | −0.036 | 0.101 * | 0 |
| | −0.014 | −0.015 | −0.05 | −0.013 | −0.017 | −0.032 | −0.031 | −0.038 | −0.031 | −0.051 | −0.038 |
| Not working | −0.024 * | 0.005 | −0.022 | 0.037 ** | 0.011 | 0.025 | −0.136 ** | −0.022 | −0.062 ** | −0.035 | 0.031 |
| | 0.01 | 0.011 | 0.039 | 0.009 | 0.012 | 0.023 | 0.022 | 0.028 | 0.022 | 0.037 | 0.027 |
| Male | −0.034 * | −0.044 ** | 0.016 | −0.078 ** | −0.067 ** | −0.105 ** | 0.031* | −0.028 | 0.064 ** | −0.047* | −0.008 |
| | 0.006 | 0.007 | 0.023 | 0.006 | 0.008 | 0.015 | 0.014 | 0.017 | 0.014 | 0.024 | 0.017 |
| Age | 0.013 * | 0.009 ** | 0.026 ** | 0.029 ** | 0.023 ** | 0.056 ** | −0.018 ** | 0.013 ** | 0 | −0.019 ** | −0.003 |
| | 0 | 0 | 0.003 | 0 | 0 | 0.001 | 0.001 | 0.002 | 0.001 | 0.003 | 0.002 |
| Married | −0.090 * | 0.071 ** | −0.304 ** | −0.177 ** | −0.097 ** | 0.088 ** | −0.085 ** | −0.203 ** | 0.034 * | −0.120 ** | −0.291 ** |
| | 0.007 | 0.008 | 0.028 | 0.007 | 0.009 | 0.018 | 0.017 | 0.021 | 0.017 | 0.028 | 0.021 |
| Years of schooling (non−Haredi) | 0.025 ** | 0.009 ** | 0.042 ** | 0.016 ** | 0.016 ** | 0.023 ** | 0.054 ** | 0.018 ** | 0.050 ** | 0.015 ** | 0.016 ** |
| | 0 | 0 | 0.002 | 0 | 0 | 0.001 | 0.001 | 0.001 | 0.001 | 0.002 | 0.002 |
| Years of schooling (Haredi) | 0.006 ** | 0.004 ** | 0 | 0.007 ** | 0.007 ** | 0.002 | 0.015 ** | 0.008 ** | 0 | 0.008* | 0.004 |
| | 0.001 | 0.001 | 0.005 | 0.001 | 0.001 | 0.002 | 0.002 | 0.003 | 0.002 | 0.004 | 0.003 |
| Center | 0.115 ** | 0.006 | 0.145 ** | 0.265 ** | 0.115 ** | 0.052 ** | 0.119 ** | 0.054 ** | 0.102 ** | −0.061 ** | 0.018 |
| | 0.006 | 0.007 | 0.022 | 0.006 | 0.007 | 0.014 | 0.013 | 0.016 | 0.013 | 0.022 | 0.016 |
| Jewish | −0.02 | −0.240 ** | 0.063 | 0.342 ** | −0.076 ** | 0.125 ** | 0.534 ** | −0.545 ** | 0 | −0.296 ** | −0.145 ** |
| | 0.011 | 0.012 | 0.045 | 0.011 | 0.014 | 0.027 | 0.026 | 0.029 | 0.025 | 0.042 | 0.03 |
| Rooms | 0.102 ** | 0.039 ** | 0.027 ** | 0.092 ** | 0.154 ** | 0.043 ** | 0.131 ** | 0.053 ** | 0.160 ** | 0.055 ** | 0.047 ** |
| | 0.002 | 0.003 | 0.009 | 0.002 | 0.003 | 0.006 | 0.006 | 0.007 | 0.006 | 0.009 | 0.007 |
| More than one car | 0.272 ** | 0.084 ** | 0.454 ** | 0.016* | 0.217 ** | 0.226 ** | 0.458 ** | 0.196 ** | 0.676 ** | 0.167 ** | 0.236 ** |
| | 0.008 | 0.009 | 0.027 | 0.008 | 0.01 | 0.019 | 0.018 | 0.021 | 0.018 | 0.029 | 0.022 |
| Log (labor income) | 0.017 ** | 0.006 ** | 0.024 ** | −0.009 ** | 0.009 ** | 0.001 | 0.034 ** | 0.022 ** | 0.069 ** | 0.008* | 0.023 ** |
| | 0.001 | 0.001 | 0.004 | 0.001 | 0.001 | 0.002 | 0.002 | 0.003 | 0.002 | 0.004 | 0.003 |
| Log (non−labor income) | 0.147 ** | 0.073 ** | 0.122 ** | 0.128 ** | 0.131 ** | 0.165 ** | 0.151 ** | 0.098 ** | 0.202 ** | 0.158 ** | 0.118 ** |
| | 0.003 | 0.003 | 0.009 | 0.002 | 0.003 | 0.006 | 0.006 | 0.007 | 0.006 | 0.01 | 0.007 |
| Cohort A | 0.110 ** | 0.044 ** | 0.203 ** | 0.178 ** | 0.159 ** | 0.326 ** | −0.022 | 0.190 ** | 0.152 ** | −0.158 ** | 0.002 |
| | 0.01 | 0.011 | 0.038 | 0.01 | 0.012 | 0.024 | 0.022 | 0.028 | 0.023 | 0.037 | 0.028 |
| Cohort B | 0.115 ** | 0.044 ** | 0.179 ** | 0.147 ** | 0.153 ** | 0.272 ** | 0.006 | 0.174 ** | 0.161 ** | −0.083* | 0.048 |
| | 0.01 | 0.011 | 0.038 | 0.01 | 0.012 | 0.024 | 0.022 | 0.028 | 0.023 | 0.037 | 0.028 |
| Cohort C | 0.080 ** | 0.034 ** | 0.135 ** | 0.112 ** | 0.110 ** | 0.168 ** | 0.013 | 0.111 ** | 0.099 ** | −0.048 | 0.046 |
| | 0.01 | 0.011 | 0.039 | 0.01 | 0.012 | 0.024 | 0.023 | 0.028 | 0.023 | 0.038 | 0.028 |
| Cohort D | 0.034 ** | 0.017 | 0.026 | 0.051 ** | 0.053 ** | 0.093 ** | −0.006 | 0.034 | 0.046* | −0.005 | −0.015 |
| | 0.01 | 0.011 | 0.039 | 0.009 | 0.012 | 0.023 | 0.022 | 0.028 | 0.022 | 0.037 | 0.027 |

**Table A3.** *Cont.*

| Explanatory Variable | Total | Food | Meals Outside of Home | Housing | Home Maintenance | Health | Education, Culture, and Entertainment | Clothing and Shoes | Transportation and Communication | Furniture and Home Equipment | Other Goods and Services |
|---|---|---|---|---|---|---|---|---|---|---|---|
| Intercept | 5.661 ** | 5.288 ** | 1.070 ** | 3.350 ** | 2.777 ** | −0.221 | 3.838 ** | 3.256 ** | 2.898 ** | 4.671 ** | 4.026 ** |
| | 0.055 | 0.061 | 0.196 | 0.053 | 0.067 | 0.129 | 0.12 | 0.148 | 0.121 | 0.198 | 0.147 |
| R−squared | 41.20% | 8.80% | 14.50% | 43.80% | 32.20% | 14.60% | 28.30% | 6.70% | 35.30% | 3.70% | 5.70% |
| Observations | 23,975 | 23,354 | 11,821 | 23,581 | 23,607 | 21,762 | 22,940 | 15,619 | 23,807 | 17,181 | 20,371 |

Notes: Standard error below coefficient. * Statistically significant at 5%. ** Statistically significant at 1%.

## Appendix D

**Table A4.** Complete results of estimating Equation (4).

| Explanatory Variable | OLS | P10 | P20 | P30 | P40 | P50 | P60 | P70 | P80 | P90 |
|---|---|---|---|---|---|---|---|---|---|---|
| Employee | 0.082 ** | 0.042 | 0.057 * | 0.079 ** | 0.078 ** | 0.077 ** | 0.081 ** | 0.109 ** | 0.109 ** | 0.099 ** |
| | 0.017 | 0.029 | 0.028 | 0.021 | 0.021 | 0.023 | 0.019 | 0.022 | 0.025 | 0.031 |
| Self−employed | 0.072 ** | 0.032 | 0.019 | 0.035 | 0.045 | 0.052 | 0.074 ** | 0.109 ** | 0.108 ** | 0.126 ** |
| | 0.02 | 0.032 | 0.034 | 0.026 | 0.024 | 0.026 | 0.021 | 0.023 | 0.027 | 0.035 |
| Not working | 0.011 | 0.016 | 0.025 | 0.013 | 0.028 | 0.035 | 0.02 | 0.017 | 0.004 | 0.006 |
| | 0.017 | 0.029 | 0.026 | 0.02 | 0.019 | 0.021 | 0.02 | 0.02 | 0.024 | 0.03 |
| Male | 0.043 ** | 0.028 | 0.025 | 0.041 | 0.032 | 0.039* | 0.049 ** | 0.071 ** | 0.063 ** | 0.051 ** |
| | 0.014 | 0.027 | 0.022 | 0.022 | 0.017 | 0.019 | 0.016 | 0.016 | 0.019 | 0.017 |
| Age | 0.084 ** | 0.004 | 0.009 | 0.059 | 0.129 ** | 0.133 ** | 0.113 ** | 0.130 ** | 0.119 ** | 0.101 * |
| | 0.026 | 0.052 | 0.036 | 0.035 | 0.043 | 0.028 | 0.025 | 0.03 | 0.028 | 0.043 |
| Married | 0.031 * | 0.014 | 0.042 | 0.028 | 0.035 | 0.038 * | 0.043 ** | 0.048 ** | 0.017 | 0.016 |
| | 0.014 | 0.027 | 0.024 | 0.021 | 0.02 | 0.018 | 0.015 | 0.016 | 0.017 | 0.02 |
| Female | −0.049 ** | −0.03 | −0.034 | −0.035 | −0.037 | −0.031 | −0.042* | −0.059 ** | −0.066 ** | −0.067 * |
| | 0.014 | 0.028 | 0.021 | 0.021 | 0.02 | 0.019 | 0.017 | 0.019 | 0.022 | 0.029 |
| Age | 0.010 ** | 0.004* | 0.007 ** | 0.009 ** | 0.009 ** | 0.010 ** | 0.011 ** | 0.013 ** | 0.014 ** | 0.014 ** |
| | 0.001 | 0.002 | 0.002 | 0.001 | 0.001 | 0.001 | 0.001 | 0.001 | 0.001 | 0.001 |
| Married | 0.070 ** | 0.118 ** | 0.119 ** | 0.109 ** | 0.106 ** | 0.082 ** | 0.066 ** | 0.051 ** | 0.039 ** | 0.014 |
| | 0.009 | 0.016 | 0.014 | 0.011 | 0.01 | 0.011 | 0.01 | 0.011 | 0.013 | 0.015 |
| Years of schooling (non−Haredi) | 0.010 ** | 0.015 ** | 0.013 ** | 0.014 ** | 0.012 ** | 0.010 ** | 0.009 ** | 0.009 ** | 0.008 ** | 0.006 ** |
| | 0.001 | 0.001 | 0.001 | 0.001 | 0.001 | 0.001 | 0.001 | 0.001 | 0.001 | 0.001 |
| Years of schooling (Haredi) | 0.005 ** | 0.003* | 0.002 | 0.001 | 0.002 | 0.004 | 0.006 ** | 0.007 ** | 0.009 ** | 0.008 ** |
| | 0.001 | 0.001 | 0.001 | 0.002 | 0.002 | 0.002 | 0.002 | 0.002 | 0.002 | 0.002 |
| Center | 0.005 | 0.017 | 0.006 | 0.014 | 0.018* | 0.011 | 0.014 | −0.003 | −0.012 | 0.007 |
| | 0.007 | 0.012 | 0.011 | 0.009 | 0.008 | 0.008 | 0.008 | 0.008 | 0.01 | 0.012 |
| Arab | −0.242 ** | −0.234 ** | −0.238 ** | −0.244 ** | −0.238 ** | −0.234 ** | −0.249 ** | −0.241 ** | −0.248 ** | −0.291 ** |
| | 0.013 | 0.026 | 0.02 | 0.018 | 0.014 | 0.017 | 0.015 | 0.015 | 0.017 | 0.024 |
| Rooms | 0.040 ** | 0.028 ** | 0.035 ** | 0.035 ** | 0.034 ** | 0.039 ** | 0.042 ** | 0.044 ** | 0.049 ** | 0.052 ** |
| | 0.003 | 0.006 | 0.004 | 0.003 | 0.004 | 0.004 | 0.003 | 0.004 | 0.004 | 0.005 |
| More than one car | 0.083 ** | 0.105 ** | 0.095 ** | 0.094 ** | 0.088 ** | 0.081 ** | 0.066 ** | 0.067 ** | 0.067 ** | 0.070 ** |
| | 0.009 | 0.017 | 0.014 | 0.012 | 0.01 | 0.011 | 0.008 | 0.01 | 0.011 | 0.014 |
| Log (labor income) | 0.006 ** | 0.013 ** | 0.013 ** | 0.007 ** | 0.007 ** | 0.007 ** | 0.006 ** | 0.003* | 0.002 | 0.002 |
| | 0.001 | 0.002 | 0.002 | 0.002 | 0.002 | 0.002 | 0.002 | 0.001 | 0.002 | 0.002 |
| Log (non−labor income) | 0.074 ** | 0.083 ** | 0.095 ** | 0.087 ** | 0.094 ** | 0.098 ** | 0.089 ** | 0.079 ** | 0.067 ** | 0.061 ** |
| | 0.003 | 0.009 | 0.009 | 0.005 | 0.004 | 0.005 | 0.006 | 0.005 | 0.006 | 0.004 |
| Cohort A | 0.047 ** | −0.004 | 0.048 * | 0.041 ** | 0.049 ** | 0.048 ** | 0.058 ** | 0.060 ** | 0.063 ** | 0.053 ** |
| | 0.012 | 0.02 | 0.021 | 0.016 | 0.014 | 0.017 | 0.014 | 0.013 | 0.016 | 0.018 |
| Cohort B | 0.047 ** | −0.002 | 0.046 * | 0.041 ** | 0.056 ** | 0.055 ** | 0.053 ** | 0.063 ** | 0.062 ** | 0.060 ** |
| | 0.012 | 0.019 | 0.02 | 0.016 | 0.015 | 0.016 | 0.013 | 0.012 | 0.015 | 0.02 |
| Cohort C | 0.037 ** | −0.021 | 0.033 | 0.028 | 0.045 ** | 0.041* | 0.043 ** | 0.046 ** | 0.050 ** | 0.055 ** |
| | 0.012 | 0.021 | 0.02 | 0.019 | 0.016 | 0.017 | 0.014 | 0.014 | 0.016 | 0.02 |
| Cohort D | 0.018 | −0.005 | 0.043* | 0.027 | 0.032* | 0.024 | 0.019 | 0.02 | 0.018 | 0.009 |
| | 0.011 | 0.019 | 0.017 | 0.016 | 0.013 | 0.014 | 0.013 | 0.013 | 0.016 | 0.018 |
| Intercept | 5.255 ** | 4.852 ** | 4.752 ** | 4.952 ** | 5.009 ** | 5.099 ** | 5.218 ** | 5.334 ** | 5.505 ** | 5.865 ** |
| | 0.062 | 0.104 | 0.102 | 0.082 | 0.076 | 0.08 | 0.071 | 0.069 | 0.092 | 0.101 |

Notes: Standard error below coefficient. * Statistically significant at 5%. ** Statistically significant at 1%.

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
