# Peer review of "Does Food Expenditure Decrease after Retirement, and for Whom?"

_sustainability, doi:10.3390/su16051992_

Round 1

Reviewer 1 Report

Comments and Suggestions for Authors

The assessed work concerns a very interesting and socially important issue, which is reducing food expenses related to retirement. This issue is part of the research on sustainable consumption (in this case, the lack of sustainable consumption was found). This issue falls under the area of interest of "Sustainability". In my opinion, the choice of this issue to study based on data from Israel is very appropriate, because the social security system in Israel is not perfect. Also a broad and at the same time very skillfully conducted review of the research results of other authors deserves recognition. The review of the literature justifies the complexity of the examined problem and the spatial diversity of results, which is related to the existence of different social security systems.

The methodological part of the work was described very thoroughly. It is important that the methodological approach (Fisher and Marchand) was indicated, which was the starting point for this research, and the changes that were introduced in it in order to expand the original methodological approach were indicated. This procedure allowed the Authors to obtain a more detailed description of the studied phenomenon. The method of successive approximations is, in my opinion, an excellent choice of the Authors. Thanks to a systematic description, it will be possible to carry out analogous studies on other populations, which will also be mutually comparable.

I also believe that the presented approach is not yet complete. In my opinion, it should take into account the fact that the decline in food consumption after retirement is also physiologically determined. It is a natural mechanism and specific to the aging process. As people age, their physical activity decreases. The demand for food of an aging person also decreases, i.e. the so-called metabolism. Please take into account that maintaining consumption at the same level at the age of 70 as at the age of 40 will lead to obesity, which has been considered an epidemic of the 21st century and leads to other diseases related to it, e.g. diabetes, atherosclerosis, etc. It is highly socially unfavorable. Perhaps my suggestion will be difficult to implement at this stage in the empirical part of the work, but this thread can be taken into account in the theoretical part. In this way, this issue can be studied in increasingly greater detail. This is not a criticism of this work, but only a suggestion to supplement it. I leave this comment to the Authors' discretion.

The literature cited in the work is sufficiently extensive and adequate to the topic. The number of self-citations is normal (not excessively high). Unfortunately, only about 29% of literature items are recent publications (from the last 5 years).

I still have a doubt about the sentence from lines 264-265: „Compared to post-retirement, ... who did not work.” Maybe I'm wrong, but please check whether it agrees with the authors' intention. In my opinion, its logical design is not perfect and could be improved.

Author Response

Thanks much for your valuable feedback. Below are your comments (those that deserved a response) and our responses.

Comment: I also believe that the presented approach is not yet complete. In my opinion, it should take into account the fact that the decline in food consumption after retirement is also physiologically determined. It is a natural mechanism and specific to the aging process. As people age, their physical activity decreases. The demand for food of an aging person also decreases, i.e. the so-called metabolism. Please take into account that maintaining consumption at the same level at the age of 70 as at the age of 40 will lead to obesity, which has been considered an epidemic of the 21st century and leads to other diseases related to it, e.g. diabetes, atherosclerosis, etc. It is highly socially unfavorable. Perhaps my suggestion will be difficult to implement at this stage in the empirical part of the work, but this thread can be taken into account in the theoretical part. In this way, this issue can be studied in increasingly greater detail. This is not a criticism of this work, but only a suggestion to supplement it. I leave this comment to the Authors' discretion.

Response: This is very much true. However, our empirical analysis controlled for age in the regression (see full regression results in the appendices). Hence, what we compare is the expenditures of two people of the same age, one retired and one still working. We recognize that this was not highlighted enough in the text and added a clarification (above table 4).

Comment: The literature cited in the work is sufficiently extensive and adequate to the topic. The number of self-citations is normal (not excessively high). Unfortunately, only about 29% of literature items are recent publications (from the last 5 years).

Response: We have added a final paragraph outlining possible future extensions based on the more recent literature.

Comment: I still have a doubt about the sentence from lines 264-265: „Compared to post-retirement, ... who did not work.” Maybe I'm wrong, but please check whether it agrees with the authors' intention. In my opinion, its logical design is not perfect and could be improved.

Response: Thank you for this comment. This indeed deserves a clarification. We added that on line 266.

Reviewer 2 Report

Comments and Suggestions for Authors

My specific comments:

(1) The authors discuss about the potential endogeneity problem in estimating the proposed model. However, the estimated model does not account for the endogeneity problem. The authors must re-estimate the model by incorporating the endogeneity problem.

(2) The authors compare the beta coefficient for the retirement period vs pre-retirement period. However, the change in consumption pattern is transitionary across the age and it is not necessarily due to retirement. The consumption at the younger age is likely to be larger than around the retirement period. If the authors introduce a dummy variable for the five-year period before the retirement, for example, and compare the coefficient for the previous periods then such transitionary effects will be apparent. The authors should control for the transitionary effects to analyse the consumption effect for the retirement.

Author Response

Thank you very much for reviewing our manuscript and providing thoughtful comments. See our responses below.

Comment: (1) The authors discuss about the potential endogeneity problem in estimating the proposed model. However, the estimated model does not account for the endogeneity problem. The authors must re-estimate the model by incorporating the endogeneity problem.

Response: Unfortunately, the data do not allow us to address the endogeneity issue. We have been very careful in the text to not claim causality and use the terminology of “consumption declines after retirement” rather than “retirement causes a decline in consumption”.

Comment: (2) The authors compare the beta coefficient for the retirement period vs pre-retirement period. However, the change in consumption pattern is transitionary across the age and it is not necessarily due to retirement. The consumption at the younger age is likely to be larger than around the retirement period. If the authors introduce a dummy variable for the five-year period before the retirement, for example, and compare the coefficient for the previous periods then such transitionary effects will be apparent. The authors should control for the transitionary effects to analyse the consumption effect for the retirement.

Response: Actually, we control for age in the regression (see full regression results in the appendices). The age coefficient turns out to be positive, in fact, but this is after controlling for other covariates. We added a clarification in the text to highlight this (above table 4).

Reviewer 3 Report

Comments and Suggestions for Authors

Thank you very much for giving me the opportunity to read the article titled "Does Food Expenditure Decrease After Retirement, and for Whom?" and making me a reviewer. While reading the text of the article, I noticed a few aspects that could improve the proposed manuscript and the discussion of the reviewed research. These are of minor significance, and I encourage the author’/’s to introduce these minor revisions.

These aspects (suggesting minor revisions) are as follows:

-  The review of the literature, used here as a research method, should be more critical and less narrative. The findings of the study are more descriptive than revealing. What theoretical contributions did your findings make to the literature? I am fairly sceptical about the results and the contribution of the study.

- Connecting to the first comment, I wonder how much we can learn from the enquiry now, when prices have changed since 2012? What can we learn from that past experience now? This is a broad topic for the discussion. 

Author Response

Comment: The review of the literature, used here as a research method, should be more critical and less narrative. The findings of the study are more descriptive than revealing. What theoretical contributions did your findings make to the literature? I am fairly sceptical about the results and the contribution of the study.

Reply: Thank you for the comment. We conducted a thorough literature review in order to describe the state-of-the-art in this field. We do not criticize the different approaches, as each one sheds additional light on the issue and each one is subject to different data constraints. Eventually, we adopted the best of all approaches to the Israeli case study of the changes in food expenditures after retirement. Following your comment, we added the following paragraph at the end of the literature review:

"To summarize, the existing literature examined the retirement-consumption puzzle from many different angles using many different empirical strategies. The literature that focused on food expenditures is much less developed, though. The purpose of this study is to adopt the most suitable empirical strategies from the general literature (and developing them further) for studying the changes in food expenditures after retirement."

We also added this sentence at the beginning of the concluding section:

"We adopted the best-practice empirical methodology from the literature on overall consumption for the specific case of food expenditures."

See also the last paragraph of the paper, which we added following the suggestion of another referee.

Comment: Connecting to the first comment, I wonder how much we can learn from the enquiry now, when prices have changed since 2012? What can we learn from that past experience now? This is a broad topic for the discussion. 

Reply: This is a good point. We have referred to the increase in the coverage of mandatory pensions in the concluding section, but have now expanded it to reflect your comment. It now reads:

"It is important to note that there were some likely changes in relevant parameters since our data were collected. Saving in pension funds is now mandatory for all workers in Israel, including the self-employed, and employment rates of older people have increased, perhaps due to improved health. Therefore, the inadequate savings explanation for the decline in food consumption after retirement is perhaps not as strong today as it has been in the past."

Round 2

Reviewer 2 Report

Comments and Suggestions for Authors

I am not convinced with revised version of the manuscript as the authors have disposed all comments and hence the reliability of the results are questionable due to the following g reasons:

(1) Endogeneity problem leads to biased estimate. Given that the estimates are likely to be biased, the causation or any nature of relationship will be misleading with the biased estimate. Hence, the findings based on the biased estimates are questionable.

(2) The inclusion of age does not control of the transitionary effects. 

Author Response

Comment: I am not convinced with revised version of the manuscript as the authors have disposed all comments...

Reply: We have not disposed all comments. We replied to the comments as best as we could. Comments made by other referees have lead to important revisions.

Comment: ...and hence the reliability of the results are questionable due to the following reasons:

(1) Endogeneity problem leads to biased estimate. Given that the estimates are likely to be biased, the causation or any nature of relationship will be misleading with the biased estimate. Hence, the findings based on the biased estimates are questionable.

Reply: We acknowledge the problem. However we have no proper instrument in the data set to correct for endogeneity bias.

(2) The inclusion of age does not control of the transitionary effects. 

Reply: we do not understand this comment. The comment from the previous round was that "The consumption at the younger age is likely to be larger than around the retirement period." We definitely agree, but this is taken care of by controlling for age. Our coefficients measure the difference in consumption between two people OF THE SAME AGE, one retired and one not.